# Circulating T cell-monocyte complexes are markers of immune perturbations

Julie G Burel[1], Mikhail Pomaznoy[1], Cecilia S Lindestam Arlehamn[1], Daniela Weiskopf[1], Ricardo da Silva Antunes[1], Yunmin Jung[2], Mariana Babor[1], Veronique Schulten[1], Gregory Seumois[1], Jason A Greenbaum[3], Sunil Premawansa[4], Gayani Premawansa[5], Ananda Wijewickrama[6], Dhammika Vidanagama[7], Bandu Gunasena[8], Rashmi Tippalagama[9†], Aruna D deSilva[1,9†], Robert H Gilman[10,11], Mayuko Saito[12], Randy Taplitz[13], Klaus Ley[2,14], Pandurangan Vijayanand[1,15], Alessandro Sette[1,15], Bjoern Peters[1,15]*

[1]Division of Vaccine Discovery, La Jolla Institute for Immunology, La Jolla, United States; [2]Division of Inflammation Biology, La Jolla Institute for Immunology, La Jolla, United States; [3]Bioinformatics core, La Jolla Institute for Immunology, La Jolla, United States; [4]Department of Zoology and Environment Sciences, Science Faculty, University of Colombo, Colombo, Sri Lanka; [5]North Colombo Teaching Hospital, Ragama, Sri Lanka; [6]National Institute of Infectious Diseases, Gothatuwa, Sri Lanka; [7]National Tuberculosis Reference Laboratory, Welisara, Sri Lanka; [8]National Hospital for Respiratory Diseases, Welisara, Sri Lanka; [9]Genetech Research Institute, Colombo, Sri Lanka; [10]Johns Hopkins School of Public Health, Baltimore, United States; [11]Universidad Peruana Cayetano Heredia, Lima, Peru; [12]Department of Virology, Tohoku University Graduate School of Medicine, Sendai, Japan; [13]Division of Infectious Diseases and Global Public Health, University of California, San Diego, La Jolla, United States; [14]Department of Bioengineering, University of California, San Diego, La Jolla, United States; [15]Department of Medicine, University of California, San Diego, La Jolla, United States

*For correspondence:
bpeters@lji.org

Present address: †Dept of Paraclinical Sciences, Faculty of Medicine, General Sir John Kotelawala Defence University, Ratmalana, Sri Lanka

Competing interests: The authors declare that no competing interests exist.

**Abstract** Our results highlight for the first time that a significant proportion of cell doublets in flow cytometry, previously believed to be the result of technical artifacts and thus ignored in data acquisition and analysis, are the result of biological interaction between immune cells. In particular, we show that cell:cell doublets pairing a T cell and a monocyte can be directly isolated from human blood, and high resolution microscopy shows polarized distribution of LFA1/ICAM1 in many doublets, suggesting in vivo formation. Intriguingly, T cell-monocyte complex frequency and phenotype fluctuate with the onset of immune perturbations such as infection or immunization, reflecting expected polarization of immune responses. Overall these data suggest that cell doublets reflecting T cell-monocyte in vivo immune interactions can be detected in human blood and that the common approach in flow cytometry to avoid studying cell:cell complexes should be re-visited.
DOI: https://doi.org/10.7554/eLife.46045.001

## Introduction

Communication between immune cells is a major component of immune responses, either directly through cell-cell contacts or indirectly through the secretion of messenger molecules such as cytokines. In particular, the physical interaction between T cells and antigen-presenting cells (APCs) is critical for the initiation of immune responses. APCs such as monocytes can take up debris from the

extracellular environment, and will display fragments of it on their surface to T cells, which can identify potentially harmful, non-self antigens. There is paucity of data regarding T cell-APCs interactions in humans in vivo, but they appear to be highly diverse in terms of structure, length and function, depending on the nature and degree of maturation of the T cell and APC (*Friedl and Storim, 2004*).

Despite the importance of interactions between immune cells, many experimental techniques in immunology specifically avoid studying cell:cell complexes, in particular for the analysis of clinical samples obtained ex vivo. The most notable example for this is in flow cytometry, in which cells are labeled with a panel of fluorochrome-conjugated antibodies, and each cell is then individually hit by a laser and its corresponding fluorescence emission spectra recorded. In this process, doublets (a pair of two cells) are routinely observed but are believed to be the results of technical artifacts due to ex vivo sample manipulation and are thus usually discarded, or ignored in data analysis (*Kudernatsch et al., 2013*).

Blood is the most readily accessible sample in humans with high immune cell content. We and others have shown circulating immune cells contain critical information that can be used for diagnostic-, prognostic- and mechanistic understanding of a given disease or immune perturbation (*Bongen et al., 2018*; *Burel et al., 2018*; *Grifoni et al., 2018*; *Roy Chowdhury et al., 2018*; *Zak et al., 2016*). Thus, whereas blood does not fully reflect what is occurring in tissues, it contains relevant information from circulating immune cells that have been either directly impacted by the perturbation, or indirectly through cell contact with tissue-resident cells in the affected compartment, including lymphoid organs.

However, the presence of dual-cell complexes (and their content) has never been studied in the peripheral blood and in the context of immune perturbations. Monocytes are a subtype of phagocytes present in high abundance in the peripheral blood, which play a critical role in both innate and adaptive immunity (*Jakubzick et al., 2017*). In particular, monocytes have the capacity to differentiate into highly specialized APCs such as macrophages or myeloid DCs (*Sprangers et al., 2016*). More recently, it has been highlighted that they might directly function as APCs and thus contribute to adaptive immune responses (*Jakubzick et al., 2017*; *Randolph et al., 2008*).

We recently identified a gene signature in memory CD4+ T cells circulating in the peripheral blood that distinguishes individuals with latent TB infection (LTBI) from uninfected individuals (*Burel et al., 2018*). Surprisingly, this dataset also led to the discovery of a group of monocyte-associated genes co-expressed in memory CD4+ T cells whose expression is highly variable across individuals. We ultimately traced this signature to a population of CD3+CD14+ cells that are not single cells but T cell:monocyte complexes present in the blood and that can be detected following immune perturbations such as disease or vaccination. The frequency and T cell phenotypes of these complexes appear to be associated with the nature of pathogen or vaccine. Thus, studying these complexes promises to provide insights into the impact of immune perturbation on APCs, T cells and their interactions.

## Results

### Unexpected detection of monocyte gene expression in CD4+ memory T cells from human subjects

We initially set out to investigate the inter-individual variability of gene expression within sorted memory CD4+ T cells from our previously characterized cohort of individuals with latent tuberculosis infection (LTBI) and uninfected controls (*Burel et al., 2018*). Within the 100 most variable genes, we identified a set of 22 genes that were highly co-expressed with each other (22-var set, *Figure 1A*, *Figure 1—source data 1*). Strikingly, many of the genes contained within the 22-var set were previously described as being highly expressed in classical monocytes (and to a lower extent non-classical monocytes) but not in T cells (*Figure 1B*, *Schmiedel et al., 2018*). In particular, the 22-var set contained the commonly used monocyte lineage marker CD14, the lysozyme LYZ and the S100 calcium binding proteins S100A8 and S100A9, which are known to be extremely abundant in monocytes (*Figure 1B*). By examining the flow cytometry data that were acquired during cell sorting and applying our memory CD4+ T cell gating strategy (*Figure 1—figure supplement 1A*), we identified that indeed there was a subpopulation within sorted memory CD4+ T cells that stained positive for CD14 (*Figure 1C*). More importantly, the proportion of memory CD4+ T cells that were CD14+ was

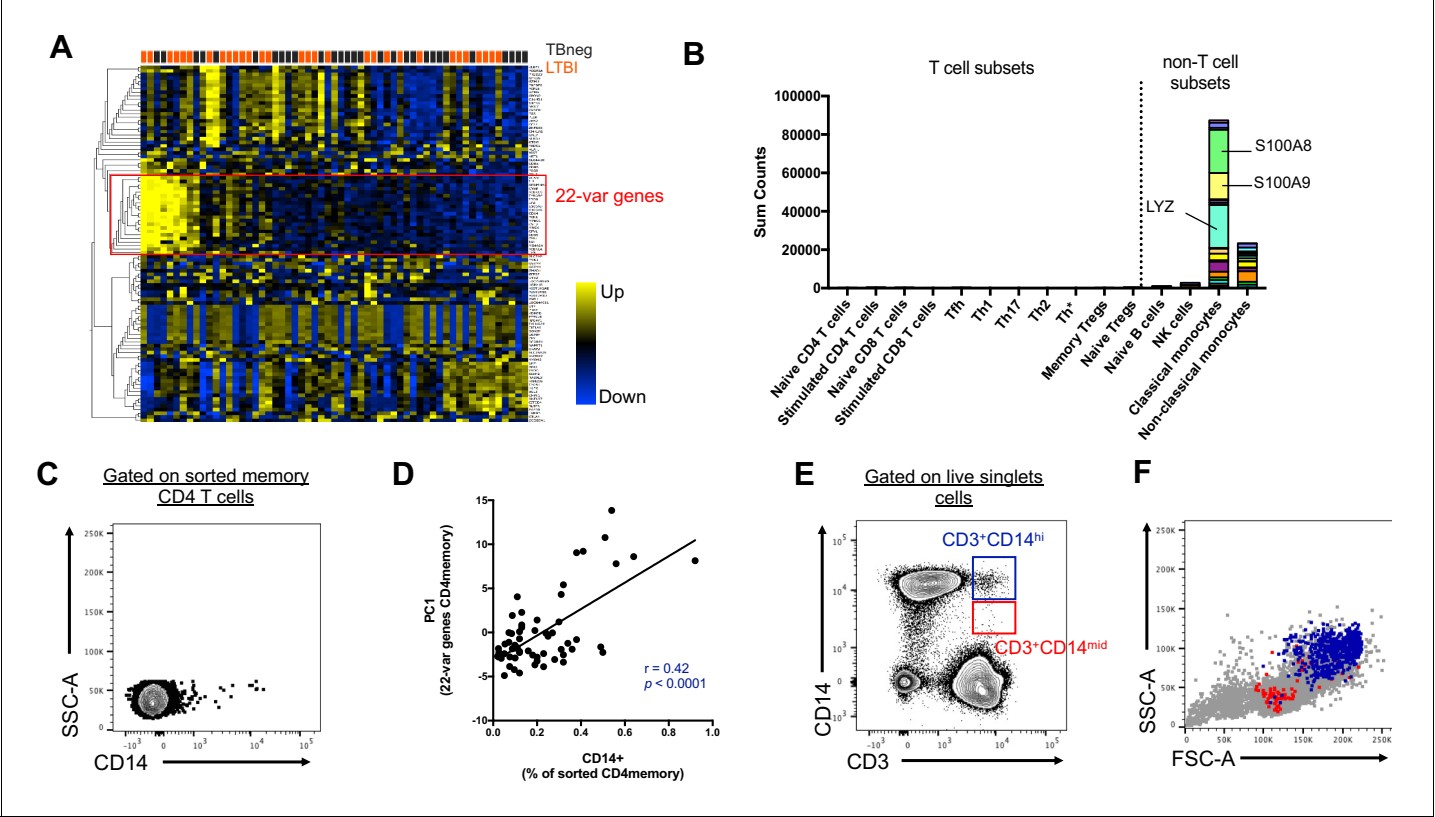

**Figure 1.** Two cell populations expressing both T cell (CD3) and monocyte (CD14) surface markers exist in the live singlet cell population of PBMC from human subjects. (**A**) The top 100 most variable genes in memory CD4+ T cells across TB uninfected (TBneg) and LTBI infected subjects. (**B**) Immune cell type specific expression of the 22-var genes identified in A). Every bar consists of stacked sub-bars showing the TPM normalized expression of every gene in corresponding cell type. Expression of genes for the blood cell types shown were taken from the DICE database (*Schmiedel et al., 2018*, http://dice-database.org/). (**C**) Detection of CD14+ events within sorted CD4+ memory T cells and (**D**) non-parametric spearman correlation between their frequency and the PC1 from the 22-var genes. (**E**) Gated on 'singlet total live cells', two populations of CD3+CD14+ cells can be identified based on the level of expression of CD14. (**F**) Based on FSC and SSC parameters, CD3+CD14hi cells are contained within the monocyte gate, whereas CD3+CD14mid cells are contained within the lymphocyte gate. Data were derived from 30 LTBI subjects and 29 TB uninfected control subjects.
DOI: https://doi.org/10.7554/eLife.46045.002

The following source data and figure supplements are available for figure 1:

**Source data 1.** Non-parametric spearman correlation between the 22-var genes in memory CD4+ T cells from human subjects.
DOI: https://doi.org/10.7554/eLife.46045.005
**Figure supplement 1.** Identification of a CD14 +population within memory CD4 T cells.
DOI: https://doi.org/10.7554/eLife.46045.003
**Figure supplement 2.** Gating strategy to identify CD3+CD14+ cells.
DOI: https://doi.org/10.7554/eLife.46045.004

positively correlated with the 22-var set expression (spearman correlation coefficient r = 0.42, p<0.0001, *Figure 1D*), suggesting that this cell subset is responsible for the expression of the monocyte-associated genes identified in *Figure 1A*. The CD14+ memory CD4+ T cell population has similar forward and side scatter (FSC/SSC) values to other memory CD4+ T cells and was thus sorted along with conventional CD14- memory CD4+ T cells (*Figure 1—figure supplement 1B*). In particular, there was no indication that CD14+ memory CD4+ T cells were the product of a technical artifact, such as dead cells or a compensation issue.

## Distinct CD3+CD14+ cell populations are present in the monocyte vs. the lymphocyte size gate

To further investigate the origin of the CD14+ T cell population, we analyzed our flow cytometry data, this time not restricting to the compartment of sorted memory T cells, but looking at all cells.

When gating on live FSC/SSC (including both monocytes and lymphocytes) singlet cells (*Figure 1—figure supplement 2*), two populations of CD3+CD14+ could be readily identified: CD3+CD14hi cells and CD3+CD14mid cells (*Figure 1E*). CD3+CD14hi cells were predominantly contained within the monocyte size gate, whereas CD3+CD14mid cells were contained within the lymphocyte size gate (*Figure 1F*).

## CD3+CD14+ cells consist of T cells bound to monocytes or monocyte debris

To better understand the nature of CD3+CD14+ cells, we aimed to visualize the distribution of their markers using imaging flow cytometry. Live events were divided into monocytes (CD3-CD14+), T cells (CD3+CD14-), CD3+CD14hi cells, and CD3+CD14mid cells (*Figure 2A*), and a random gallery of images was captured for each population. As expected, monocytes and T cells contained exclusively single cells that expressed either CD14 (monocytes) or CD3 (T cells), respectively (*Figure 2B*, *first and second panel*). To our surprise, CD3+CD14hi cells contained predominantly two cells, sometimes even three cells, but no single cells (*Figure 2B*, *third panel*). The doublets (or triplets) always contained at least one CD14+ cell, and one CD3+ cell (*Figure 2B*, *third panel*). CD3+CD14mid cells contained predominantly single CD3+ cells, but also some doublets of one CD3+ cell and one CD14 + cell, but with CD14 expression lower than average monocytes (*Figure 2B*, *fourth panel*). The majority of CD3+ T cell singlets in the CD3+CD14mid population, but not in the CD3+CD14 T cell population, contained CD14+ particles, often seen at the periphery of the CD3+ T cell membrane (*Figure 2B–C*). Looking more closely at the CD14+ particles contained within the CD3+ CD14 mid population using confocal microscopy, they were found to have size and shape similar to cell debris (*Figure 2D*). To confirm our initial observation, we repeated the experiment with multiple individuals, and compared for each cell population the aspect ratio and area from the brightfield parameter collected with the image stream. Doublets are known to present a larger area but reduced aspect ratio, when compared to single cells. Thus, their overall ratio (area vs aspect ratio) is greater than in single cells. As expected, the area vs aspect ratio was significantly higher for CD3+CD14hi cells and CD3+CD14mid cells compared to single monocytes and T cells, and events in these two cell populations were found predominantly in the 'doublet gate' (*Figure 2E–F*). CD3+CD14hi cells also had a significantly higher ratio compared to CD3+CD14mid cells (*Figure 2F*).

Taken together, these results demonstrate that CD3+CD14hi cells are tightly bound T cell:monocyte complexes, in such strong interaction that sample processing and flow cytometry acquisition did not break them apart. Conversely, the CD3+CD14mid population appears to predominantly consist of single CD3+ T cells with attached CD14+ cell debris. This conclusion is further supported by CD3+CD14hi complexes being found in the monocyte size gate, whereas CD3+CD14mid cells were falling into the lymphocyte size gate (*Figure 1F*).

For the remaining of the manuscript, we refer to T cell:monocyte complexes as the CD3+CD14 + population gated from live singlets cells, as represented in *Figure 1—figure supplement 2*.

## T cell:monocyte complexes are not the result of ex vivo sample manipulation

Next, we sought to determine whether the physical association of T cells and monocytes within the T cell:monocyte complexes was the result of random cellular proximity or non-specific antibody staining during ex vivo sample manipulation, or if the complexes are originally present in peripheral blood. We could readily detect T cell:monocyte complexes in freshly isolated PBMC, and at similar frequencies as the same samples after cryopreservation (*Figure 2G*). In another set of samples, using red blood cell (RBC) magnetic depletion (and thus minimal sample manipulation), we could successfully identify T cell:monocyte complexes directly from whole blood at frequencies matching the same sample after PBMC isolation (*Figure 2—figure supplement 1A*). We also assessed the effect of the anti-coagulant used for blood collection, and found similar frequencies of T cell:monocyte complexes using either heparin or EDTA, in both fresh and frozen PBMC (*Figure 2—figure supplement 1B*). More strikingly, in a small healthy population bled longitudinally one week apart, their frequency was variable between individuals, but highly stable over time within each individual (non-parametric spearman correlation r = 1 and r = 0.9 for fresh and frozen PBMC, respectively, *Figure 2—figure supplement 1C*). Finally, to rule out that non-specific binding of antibody conjugates

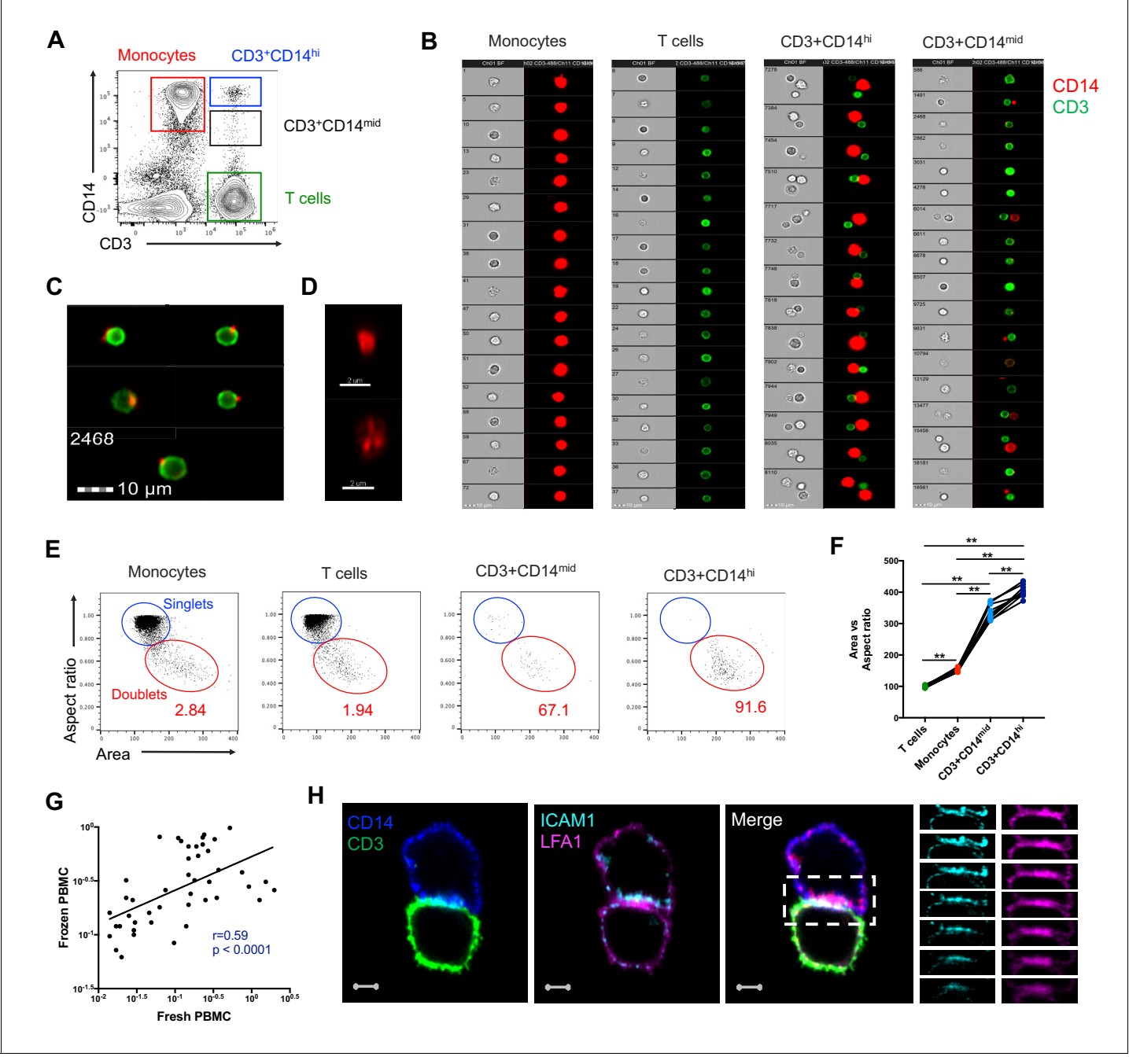

**Figure 2.** CD3+CD14+ cells are tightly bound T cell:monocyte complexes that represent in vivo association. (**A**) Gating strategy and (**B**) random gallery of events for monocytes (CD14+CD3), T cells (CD3+CD14-), CD3+CD14hi cells and CD3+CD14mid cells determined by imaging flow cytometry (ImageStreamX, MkII Amnis Amnis). CD14+ cell debris were identified within CD3+CD14mid cells (**C**) by imaging flow cytometry and (**D**) confocal microscopy after bulk population cell sorting. (**E**) Plots and (**F**) Ratio of Aspect ratio vs Area of the brightfield parameter for monocytes (CD14+CD3-), T cells (CD3+CD14-), CD3+CD14hi cells and CD3+ CD14 mid cells, determined by imaging flow cytometry. (**G**) Non-parametric Spearman correlation of the frequency of T cell:monocyte complexes in paired fresh PBMC vs cryopreserved PBMC derived from 45 blood draws of healthy subjects. T cell: monocyte complexes were defined as the CD3+CD14+ cell population gated from live singlets as represented in *Figure 1—figure supplement 2*. (**H**) Single z-plan (0 μm) images (*left*) and z-plane stacks (*right*) of the region marked (dashed rectangle) from one sorted CD3+ CD14+ T cell:monocyte complex displaying accumulation of LFA1 and ICAM1 at the interface. Images show expression of CD14 (blue), CD3 (green), ICAM1(Cyan), and LFA1 (Magenta). Relative z-positions are indicated on the right, and scale bars represent 2 μm. Imaging flow cytometry data was derived from 10 subjects across three independent experiments and microscopy data was representative of the analysis of n = 105 CD3+CD14+ complexes isolated from three subjects across three independent experiments.

DOI: https://doi.org/10.7554/eLife.46045.006

*Figure 2 continued on next page*

*Figure 2 continued*

The following figure supplements are available for figure 2:

**Figure supplement 1.** Technical variations in sample preparation do not impact the frequency of T cell:monocyte complexes.
DOI: https://doi.org/10.7554/eLife.46045.007
**Figure supplement 2.** Accumulation of CD3, LFA1 and ICAM1 at the interface of a T cell:monocyte complex.
DOI: https://doi.org/10.7554/eLife.46045.008
**Figure supplement 3.** Effect of physical and chemical sample manipulation on the frequency of T cell:monocyte complexes.
DOI: https://doi.org/10.7554/eLife.46045.009

used for flow cytometry staining could be the result of complex formation (via Fc-receptors, known to be highly abundant in the surface of monocytes), we compared the frequency of T cell:monocyte complexes obtained from cryopreserved PBMC in the presence or absence of a Fc-receptor blocking reagent. The frequency of T cell:monocyte complexes frequencies was unchanged when blocking Fc-receptor binding of conjugated antibodies (*Figure 2—figure supplement 1D*).

Taken together these data rule out that ex vivo manipulation of the blood sample could be responsible for T cell:monocyte complexes formation and thus suggest their presence in vivo in peripheral blood.

## T cell:monocyte complexes show increased expression of adhesion molecules at their interface

During T cell recognition of epitopes on APCs such as monocytes, the two cells are known to form an 'immune synapse' at their contact point, which is stabilized by key adhesion molecules such as LFA1 on the T cell, and ICAM1 on the APC (*Dustin, 2014*). Upon interaction these two molecules undergo a drastic redistribution by focusing almost exclusively at the cell:cell point of contact, thus forming a 'ring' that can be visualized (*Wabnitz and Samstag, 2016*). To identify candidate immunological synapses in T cell:monocyte complexes, we used high resolution Airyscan images of sorted doublets (see *Figure 1—figure supplement 2* for sorting strategy). Almost a third (thirty out of 105, 29%) of doublets analyzed from three different individuals displayed accumulation and polarization of ICAM1 and LFA1 at their interfaces (*Figure 2H*). The percentage of polarized doublets ranged from 17% to 67% between the subjects. In seven doublets, CD3 also accumulated together with LFA1 (*Figure 2—figure supplement 2*). However, we did not find well developed, classical immunological synapses, defined by central accumulation of CD3 and LFA1 exclusion from central region of a synapse (*Monks et al., 1998*; *Thauland and Parker, 2010*). Overall, this suggests that a significant fraction of the detected T cell:monocyte complexes utilizes adhesion markers associated with T cell: APC synapse formation to stabilize their interaction, but they do not appear to undergo active TCR signaling at the moment of capture and acquisition.

To assess the strength of the adhesion between the T cell and the monocyte within complexes, we explored various in vitro conditions in which we attempted to break down their interaction. As seen in *Figure 2—figure supplement 3*, in all three individuals tested, the frequency of T cell:monocyte complexes was reduced after vigorous pipetting up/down, and the strongest 'destructive effect' was observed with mild sonication. Incubation with RBC lysis buffer disrupted the T cell:monocyte complexes in one out of three individuals, while addition of high concentration of anti-chelating agent EDTA had no effect on their frequency. Thus, it appears that it is possible to disrupt T cell: monocyte complexes with physical methods. Conversely, we tried to promote their in vitro formation by stimulating PBMC for two hours to several days, with various concentrations of highly antigenic stimuli (such as LPS, PHA, SEB or live BCG) without success, which might be due to issues with monocytes attaching to the culture plates (data not shown).

## Conventional flow cytometry parameters could not differentiate between T cells and monocytes in a complex versus not in a complex

So far, our identification of T cell:monocyte complexes has been solely relying on the co-expression of CD3 and CD14 within cells falling into the live singlet gate (*Figure 1—figure supplement 2*). In order to fine-tune their detection, we investigated whether we could identify some additional flow cytometry parameters that could separate T cells and monocytes in a complex vs. not in a complex.

The area (A), height (H) and width (W) of the peak from forward and side scatter parameters are routinely used in flow cytometry to identify doublets. A non-linear staining between any 2D combination of these three parameters is indicative of cell aggregates. The density plot of CD3+CD14+ cells overlapped with both CD3+ cells and especially CD14+ cells for any combination of A, H and W for both FSC and SSC parameters (*Figure 3A*). An additional parameter often used to 'clean up' the gating of cells within biological samples is the use of CD45-SSC gating (*Harrington et al., 2012*). We found that the frequency of T cell:monocyte complexes was not affected by applying an initial CD45-SSC gate filtering (*Figure 3B*, see gating in *Figure 3—figure supplement 1*). In conclusion, we could not find any parameter from conventional flow cytometry that could separate singlets and T cell:monocyte complexes with sufficient resolution.

## No difference in T cell and monocyte canonical marker expression was identified between T cells and monocytes in a complex versus not in a complex

In parallel, we analyzed the expression of various markers known to be exclusively expressed by either monocytes or T cells within T cell:monocyte complexes from healthy individuals. Monocytes in a complex (CD14+CD3+) vs. not in a complex (CD14+CD3-) had similar expression for monocyte canonical markers: CD33, CD36, CD64 and CD163 (*Figure 3C*). Similarly, no difference in the levels of expression of T cell canonical markers, CD2, CD5, CD7 and CD27 was observed between T cells in a complex (CD3+CD14+) vs. not in a complex (CD3+CD14-) (*Figure 3D*). Additionally, both CD4 and CD8 T cell subsets could be found in association with a monocyte (*Figure 3E*), as well as naïve and memory phenotypes (*Figure 3F*). Thus, there was no obvious T cell or monocyte marker that could differentiate between monocytes and T cells present in a complex vs. not in a complex, suggesting the ability to form complexes is a general property of all CD14 +monocytes and T cell subsets.

## The frequency of T cell:monocyte complexes varies in the context of diverse immune perturbations

Next, we thought to examine whether the formation of T cell:monocyte complexes is affected by immune perturbations. In order to accurately assess and compare the frequency of complexes between cells of different types across different donor cohorts, we need to consider that their frequency is dependent on the abundance of its two components. Indeed, in healthy subjects, where we expect constant affinity between T cells and monocytes, we observed that the frequency of CD3 +CD14+ cells is a linear function of the product of singlet monocyte and T cell frequencies (*Figure 4A*). To correct for this, we elected to express the abundance of T cell:monocytes complexes as a constant of association Ka, where similarly to a constant of chemical complex association, the frequency of T cell:monocyte complexes is divided by the product of the frequency of both T cells and monocytes (*Figure 4B*). As T cell and monocyte frequencies in the blood can fluctuate greatly during immune perturbations, the Ka is a more accurate readout of the likelihood of T cell:monocyte complex formation as opposed to raw frequencies, the latter being biased towards the overall abundance of each subset forming the complex.

We first investigated the T cell:monocyte Ka in the context of two diseases where monocytes are known to be important, namely active tuberculosis (TB) infection and dengue fever. In the case of TB, although macrophages are known to be the primary target for *Mycobacterium tuberculosis* (Mtb) infection and replication, monocytes can also be infected and contribute to the inflammatory response (*Srivastava et al., 2014*). In active TB subjects, we found a significant decrease in T cell:monocyte Ka at 2 months post treatment (*Figure 4C*). At the time of diagnosis, some subjects displayed a Ka much higher than any uninfected or LTBI individuals, but because of the high heterogeneity within the active TB cohort, these differences did not reach statistical significance (*Figure 4—figure supplement 1*). Dengue virus predominantly infects monocytes in the peripheral blood (*Kou et al., 2008*), and circulating monocyte infection and activation is increased in dengue hemorrhagic fever (the more severe form of dengue fever) (*Durbin et al., 2008*). In subjects with acute dengue fever from Sri Lanka, patients that developed hemorrhagic fever had higher T cell:monocyte Ka upon hospitalization compared to healthy, previously infected subjects (blood bank donors seropositive for dengue antibodies) (*Figure 4D*). In contrast, patients with a less severe form of acute

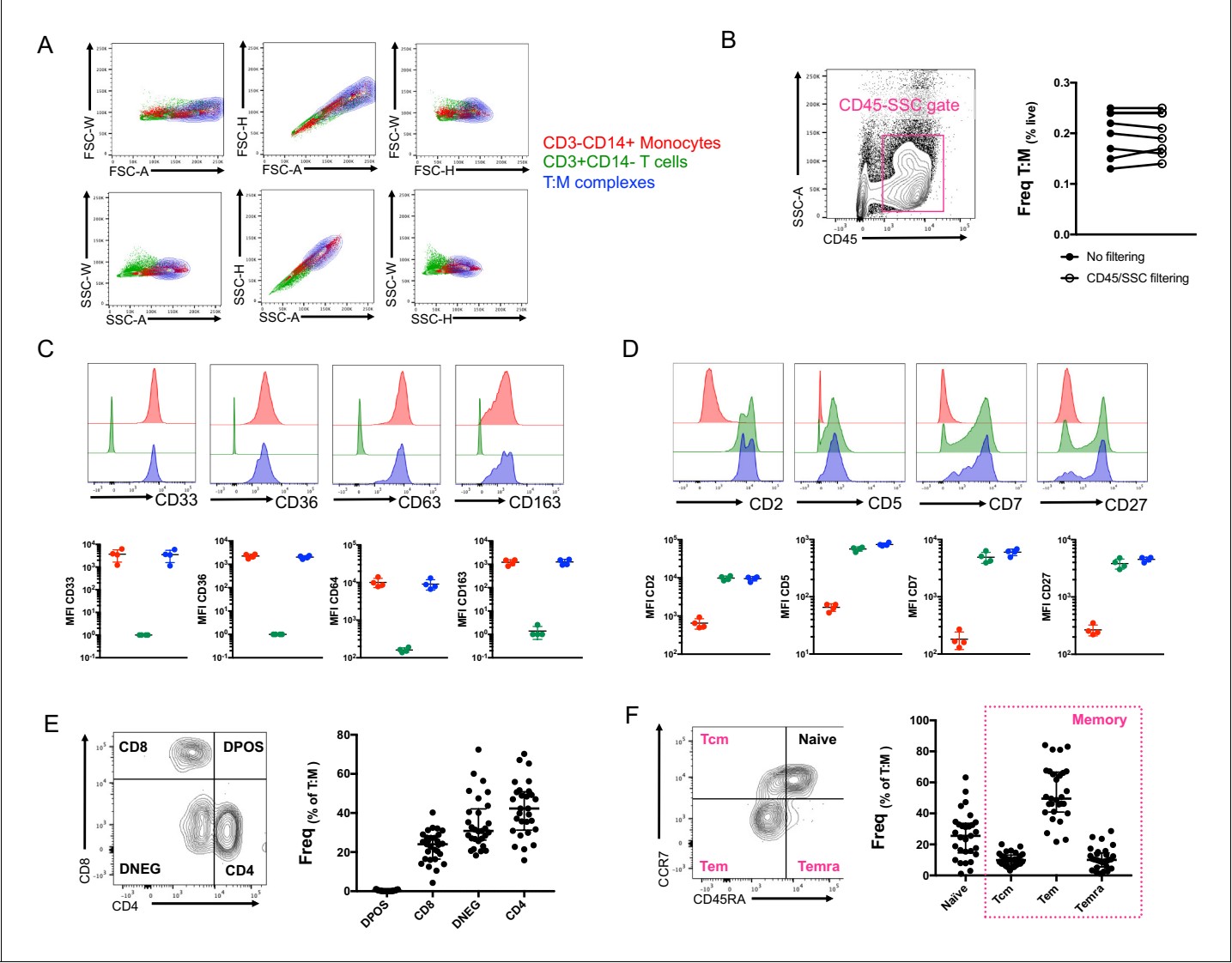

**Figure 3.** Conventional flow cytometry parameters and expression of T cell/monocyte canonical markers cannot differentiate between T cells and monocytes in a complex vs. not in a complex. (A) 2D density plots of A, H and W from FSC and SSC parameters for CD3-CD14+ Monocytes (red), CD3+CD14- T cells (green) and CD3+CD14+ T cell:monocyte complexes (T:M, blue). Representative staining of one healthy individual. (B) Frequency of T cell:monocyte complexes cells with or without addition of CD45-SSC filtering gate (see Figure 3 – figure supplement 1 for gating strategy). Expression of canonical markers for (C) monocytes and (D) T cells in CD3-CD14+ Monocytes (red), CD3+CD14- T cells (green) and CD3+CD14+ T cell:monocyte complexes (T:M, blue). (E) Expression of CD4 and CD8 and division into T cell subsets within T cell:monocyte complexes. (F) Expression of CD45RA and CCR7 and division into naïve, central memory (Tcm), effector memory (Tem) and effector memory re-expressing CD45RA (Temra) subsets within T cell: monocyte complexes. Data derived from frozen PBMC of n=30 (A, E, F), n=8 (B) and n=4 (C, D) healthy individuals. Unless otherwise stated, T cell: monocyte complexes were defined as the CD3+CD14+ cell population gated from live singlets as represented in *Figure 1—figure supplement 2*.
DOI: https://doi.org/10.7554/eLife.46045.010

The following figure supplement is available for figure 3:

**Figure supplement 1.** Gating strategy to identify CD3+CD14+ cells with or without a CD45-SSC gate filtering.
DOI: https://doi.org/10.7554/eLife.46045.011

dengue infection showed no significant difference in T cell:monocyte Ka compared to healthy, previously infected donors (*Figure 4D*).

To assess whether vaccination also impacted the formation of T cell:monocyte complexes, we obtained samples from healthy adults that received the tetanus, diphtheria and pertussis (Tdap) booster vaccination. We indeed observed a significantly higher T cell:monocyte Ka at three days

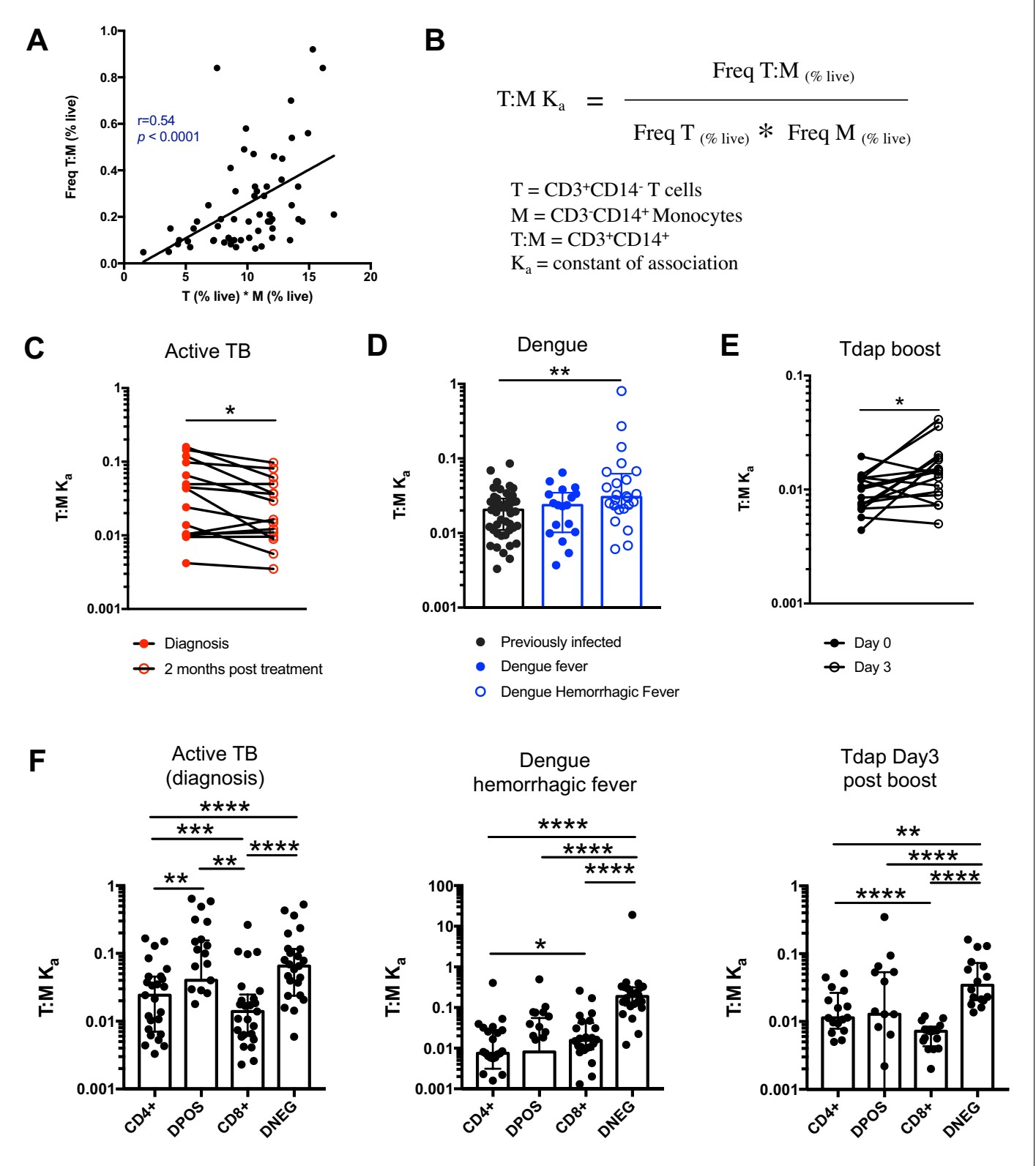

**Figure 4.** The constant of association Ka between monocytes and T cells (and T cell subsets) varies with the presence and nature of immune perturbations. (A) Non-parametric spearman correlation between the frequency of T cell:monocyte complexes and the product of singlet T cells and monocyte frequencies in healthy subjects (n = 59). (B) Formula for the calculation of the T cell:monocyte constant of association Ka. T cell:monocyte complexes constant of association Ka in (C) active TB subjects at diagnosis and 2 months post treatment (n = 15), (D) individuals with acute dengue

*Figure 4 continued on next page*

*Figure 4 continued*

fever (n = 18), acute dengue hemorrhagic fever (n = 24) or previously infected (n = 47) and (E) previously vaccinated healthy adults (n = 16) before and three days post boost with Tdap vaccine, calculated as explained in B). (F) The constant of association Ka between monocytes and T cell subsets in active TB subjects at diagnosis (n = 25), individuals with acute dengue hemorrhagic fever (n = 24) and previously vaccinated healthy adults three days post boost with Tdap vaccine (n = 16), calculated as explained in B). Statistical differences over time and across cell populations within subjects were determined using the non-parametric paired Wilcoxon test; other statistical differences were determined using the non-parametric Mann-Whitney test; *, p<0.05; **, p<0.01; ***, p<0.01; ****, p<0.0001. Plots represent individual data points, median and interquartile range across all subjects within each cohort. Raw frequencies of T cell:monocyte complexes for the different disease cohorts are available on Figure 4—figure supplement 4. T cell:monocyte complexes were defined as the CD3+CD14+ cell population gated from live singlets as represented in *Figure 1—figure supplement 2*. CD4 and CD8 subsets within T cell:monocyte complexes were defined as presented in *Figure 3E*.

DOI: https://doi.org/10.7554/eLife.46045.012

The following figure supplements are available for figure 4:

**Figure supplement 1.** T cell:monocyte constant of association Ka in subjects with active TB, latent TB or TB uninfected individuals.
DOI: https://doi.org/10.7554/eLife.46045.013

**Figure supplement 2.** T cell:monocyte constant of association Ka fluctuates as a function of time following Tdap boost administration.
DOI: https://doi.org/10.7554/eLife.46045.014

**Figure supplement 3.** Comparison of constant of association Ka between monocytes and T cell subsets across different immune perturbations.
DOI: https://doi.org/10.7554/eLife.46045.015

**Figure supplement 4.** Frequencies of T cell:monocyte complexes in different immune perturbation models.
DOI: https://doi.org/10.7554/eLife.46045.016

post boost compared to baseline (*Figure 4E*), but no significant changes at one, seven or fourteen days post boost (*Figure 4—figure supplement 2*). Taken together, these data confirm that circulating T cell:monocyte complexes can be found directly ex vivo in different immune perturbations, and their likelihood of formation is associated with clinical parameters such as disease severity, and they fluctuate as a function of time post treatment and post vaccination.

## T cells with different phenotypes are found in T cell:monocyte complexes dependent on the nature of the immune perturbation

Finally, we reasoned that if immune perturbations increase the formation of T cell:monocyte complexes, then the nature of the T cells contained in the complexes could provide insights into which T cells are actively communicating with monocytes in vivo. In particular, the T cell subsets that will associate with an APC for the different perturbations studied above are expected to be distinct, and thus their likelihood to form a complex with a monocyte might differ too. The Tdap vaccine contains exclusively protein antigens and is known to elicit predominantly memory CD4+ T cell responses (*da Silva Antunes et al., 2018*). Mtb is a bacterial pathogen known to trigger strong CD4 + responses (*Lindestam Arlehamn et al., 2016*) as opposed to dengue virus, which is a viral antigen and thus expected to elicit CD8+ responses.

Similarly to global T cell:monocyte complexes (*Figure 4C–E*), we calculated the constant of association Ka with monocytes for each CD4/CD8 T cell subset individually. In subjects with active TB, the Ka between monocytes and CD4+CD8+ (DPOS) T cells or CD4+ T cells was significantly higher than for CD8+ T cells (*Figure 4F*) and both DPOS and CD4+ T cell:monocyte complexes had higher Ka in active TB compared to dengue hemorrhagic fever (*Figure 4—figure supplement 3*). Dengue hemorrhagic fever showed a higher T cell:monocyte Ka for CD8+ over CD4+ cells whereas Tdap day three post boost showed the opposite, with highest Ka for CD4+ over CD8+ cells (*Figure 4F*). The CD8+ T cell:monocyte Ka was also higher in Dengue and active TB compared to Tdap boost (*Figure 4—figure supplement 3*). Thus, the magnitude of Ka in CD4+ vs CD8+ T cell subsets matched what is expected based on the nature of immune perturbation. Interestingly, for all three immune perturbations studied the highest Ka with monocytes across all T cell subsets was for CD4-CD8- (DNEG) T cells (*Figure 4F*), and this effect was most pronounced in dengue (*Figure 4—figure supplement 3*). These cells could constitute gamma-delta T cells that are known to be strongly activated in the peripheral blood during acute dengue fever (*Tsai et al., 2015*).

In summary, these data indicate that the T cell subsets that are preferentially associated with monocytes differ from their individual frequencies in PBMC, and follow different patterns in the three

systems studied, further supporting the notion that these complexes are not the result of random association, and are specific to the nature of the immune perturbation.

## Discussion

The unexpected detection of monocyte genes expressed in cells sorted for memory T cell markers led to the discovery that a population of CD3+CD14+ cells exist within the 'live singlet' events gate and that these cells are T cells that are tightly associated with monocytes, and less frequently, with monocyte-derived debris. Their presence in freshly isolated cells and the fact that a significant fraction of the complexes showed enriched expression for LFA1/ICAM1 adhesion molecules at their interface, suggest that they are not the product of random association of cells during processing, but represent interactions that occurred in vivo prior to the blood draw. The frequency of T cell: monocyte complexes fluctuated over time in the onset of immune perturbations such as following TB treatment or Tdap boost immunization and correlated with clinical parameters such as disease severity in the case of dengue fever. Furthermore, the T cell subset in preferential association within the monocyte in a complex varies in function of the nature of the immune perturbation.

Our initial observation of the presence of monocyte genes within the transcriptome of T cells was focused on memory CD4 T cells. This cell population was elected since our study aimed to define novel immune signatures associated with Mtb-specific CD4 T cells (*Burel et al., 2018*), which are expected to almost exclusively fall into the memory compartment in the context of latent TB infection (*Lindestam Arlehamn et al., 2013*). However, since we have found that both CD4 and CD8 T cells, from both memory and naïve phenotype, can be found in a complex with a monocyte, we think similar results would have been obtained with other sorted T cell populations. We have since detected the expression of monocytes-associated genes in several other T cell subsets, including memory CD8 T cells, and total CD4 T cells (unpublished observations).

Intact T cell:monocyte complexes were almost exclusively found in the top area of the FSC/SSC 2D plot, and were associated with high CD14 expression. In contrast, T cells with monocyte debris were associated with FSC/SSC values similar to regular non-complex T cells and an intermediate CD14 expression. This cell population might be the result of T cell:monocyte complexes from which the monocyte was disrupted during sample preparation or flow cytometry acquisition. Alternatively, these CD3+CD14mid cells could be the result of plasma membrane fragments exchange from monocytes to T cells following interaction. This phenomenon, known as trogocytosis, has been described to occur during cellular encounters between several immune cell types, including monocytes and T cells (*Daubeuf et al., 2010*; *HoWangYin et al., 2011*).

Taken together, our results suggest circulating CD3+CD14+ complexes appear to be the result of in vivo interaction between T cells and monocytes. The origin and location of the complexes' formation is still unknown. These interactions might be occurring directly in the blood. Alternatively, it is possible that T cell:monocyte complex formation does not initially occur in peripheral blood, but rather in tissues or draining lymph nodes, and these complexes are then excavated into the peripheral circulation. The most studied physical interaction between T cells and monocytes is the formation of immune synapses. We found that about a third of complexes displayed LFA1/ICAM1 mediated interaction similarly to immune synapses, but no CD3 polarization. The immune synapse formation is a highly diverse event in terms of length and structure (*Friedl and Storim, 2004*), so it is possible that not all detected complexes are at the same stage in the interaction. In some complexes, the nature (and structure) of the architectural molecules forming the cell:cell contact might differ from traditional immune synapses, too. Studying the nature and physical properties of these interactions could provide insights into how T cells and monocytes can physically interact. Additionally, because monocytes are not the only cell type known to associate with T cells, we think the ability to form complexes with T cells should not be restricted to monocytes, but could apply more broadly to any APC. Thus, it is possible that other types of complexes pairing a T cell and other APCs such as B cells or dendritic cells can be found in the peripheral blood.

Increased immune cell:cell interactions might not necessarily always correlate with onset of immune perturbations. Nevertheless, our preliminary data suggest that determining the constant of association Ka of the T cell:monocyte (and likely more broadly any T cell:APC) complexes can indicate the presence of an immune perturbation to both clinicians and immunologists. We think that the Ka, rather than the frequency of live cells, is a more relevant parameter to measure the

occurrence of T cell:monocyte complexes (and thus the in vivo affinity between T cells and monocytes) since it corrects for random/aleatory association that is directly dependent on the T cell subset or monocyte abundance. For instance, DNEG T cells are in lower abundance than CD4+ or CD8+ T cells in peripheral blood, and thus DNEG T cells in a complex with a monocyte are found at a lower frequency compared to CD4+ or CD8+ T cell:monocyte complexes (*Figure 4—figure supplement 4D*). However, interestingly, their Ka is consistently much higher than CD4+ or CD8+ T cells across all disease cohorts analyzed (*Figure 4F*), suggesting a higher affinity of DNEG cells to monocytes. This information would have been missed if only frequencies were considered. In dengue infected subjects, a higher T cell:monocyte Ka at time of admission was associated with dengue hemorrhagic fever, the more severe form of disease. The distinction between hemorrhagic vs. non-hemorrhagic fever may become clear only days into hospitalization, so the ability to discriminate these two groups of individuals at the time of admission has potential diagnostic value. In the case of active TB, subjects presented a very high variability at diagnosis that might reflect the diverse spectrum associated with the disease (*Pai et al., 2016*), but for all subjects a significant decrease in T cell:monocyte Ka was observed upon treatment. This could thus be a tool to monitor treatment success and predict potential relapses. It will of course be necessary to run prospective trials to irrefutably demonstrate that the likelihood of association between T cells and monocytes have predictive power with regard to dengue disease severity or over the course of TB treatment. Additionally, the T cell:monocyte Ka was increased three days following Tdap booster vaccination. Therefore, in vaccine trials, it could be examined as an early readout to gage how well the immune system has responded to the vaccine. Finally, in apparently 'healthy' populations, or those with diffuse symptoms, an unusually high T cell:monocyte Ka in an individual could be used as an indicator of a yet to be determined immune perturbation.

Beyond detecting abnormal frequencies of T cell:monocyte complexes, characterizing the T cells and monocytes in these complexes might provide insights into the nature of immune perturbation and subsequent immune response based on which complexes were formed. Our data suggest that there are drastic differences in terms of T cell subsets in the complexes. As aforementioned, despite their lower frequency over CD4+ and CD8+ T cells in the peripheral blood, DNEG T cells show a clear increased association with monocytes. Gamma-delta T cells constitute the majority of circulating DNEG T cells in humans, and LFA1 dependent crosstalk between gamma-delta T cells and monocytes has been shown to be important in the context of bacterial infections (*Eberl et al., 2009*), which might be also generalized to viral infections. Thus, the DNEG T cell:monocyte complexes might well represent a novel type of interaction between T cells and monocytes, not necessarily involving classical alpha-beta T cells or involving the formation of 'traditional' immune synapses. Aside from the enrichment for DNEG cells in T cell:monocyte complexes, we also observed a relatively high Ka for DPOS cells in T cell:monocyte complexes in all samples analyzed, despite their very low abundance amongst T cells. Circulating DPOS T cells have been described in the context of several infections, in particular from viruses. They are associated with enhanced effector functions such as proliferation, cytotoxicity and cytokine production (*Kitchen et al., 2004*; *Nascimbeni et al., 2004*). DPOS cells might thus represent a specific subset of T cells with enhanced surveillance and cell:cell communication functions, and thus have higher affinity for APCs hence higher likelihood to be found in a complex with a monocyte. Finally, we also found that the CD4 vs CD8 phenotype of the T cell present in complexes depends on the nature of the immune perturbation studied, and reflects the expected polarization of immune responses. Thus, looking for additional characteristics from T cells and monocytes present in the complexes, such as the expression of tissue homing markers, specific TCRs and their transcriptomic profile might provide further information about the fundamental mechanisms underlying immune responses to a specific perturbation.

Why were T cell:monocyte complexes not detected and excluded in flow cytometry based on gating strategies to avoid doublets? Surprisingly, all usual parameters (pulse Area (A), Height (H) and Width (W) from forward and side scatter) looked identical between T cell:monocyte complexes and singlet T cells or monocytes. The only parameter that could readily distinguish between intact CD3+CD14hi complexes and single T cells or monocytes was the brightfield area parameter from the imaging flow cytometer, which is a feature absent in non-imaging flow cytometry. Thus, it seems that gating approaches and parameters available in conventional flow cytometry are not sufficient to completely discriminate tightly bound cell pairs from individual cells.

Given that T cell:monocyte complexes are not excluded by conventional flow cytometry gating strategies, why were they not reported previously? Examining our own past studies, a major reason is that lineage markers for T cells (CD3), B cells (CD19) and monocytes (CD14) are routinely used to remove cells not of interest in a given experiment by adding them to a 'dump channel'. For example, most of our CD4+ T cell studies have CD8, CD19 and CD14, and dead cell markers combined in the same channel (*Arlehamn et al., 2014*; *Burel et al., 2018*). Other groups studying for example CD14 + monocytes are likely to add CD3 to their dump channel. This means that complexes of cells that have two conflicting lineage markers such as CD3 and CD14 will often be removed from datasets early in the gating strategy. Additionally, the detection of complexes by flow cytometry is not straightforward. In our hands, we have found that conventional flow analyzers give low frequency of complexes and poor reproducibility in repeat runs. This is opposed to cell sorters, presumably due to differences in their fluidics systems, which puts less stress on cells and does not disrupt complexes as much. Both the routine exclusion of cell populations positive for two conflicting lineage markers and the challenges to reproduce such cell populations on different platforms has likely contributed to them not being reported.

Moreover, even if a panel allows for the detection of complexes, and there is a stable assay used to show their presence, there is an assumption in the field that detection of complexes is a result of experimental artifacts. For example, we found a report of double positive CD3+CD34+ cells detected by flow cytometry in human bone marrow, which followed up this finding and found them to be doublets using microscopy imaging. The authors concluded that these complexes are the product of random association and should be ignored (*Kudernatsch et al., 2013*). Their conclusion may well be true for their study, but it highlights a common conception in the field of cytometry that pairs of cells have to be artifacts. Another study described CD3+CD20+ singlets cells observed by flow cytometry as doublets of T cells and B cells, and also concluded them to be a technical artifact, in the sense that these cells are not singlets double expressing CD3 and CD20 (*Henry et al., 2010*). In this case however, authors pointed out that 'Whether the formation of these doublets is an artifact occurring during staining or is a physiologic process remains to be determined' (*Henry et al., 2010*).

We ourselves assumed for a long time that we might have an artifact finding, but given the persistent association of T cell:monocyte complexes frequency and phenotype with clinically and physiologically relevant parameters, we came to a new conclusion: cells are meant to interact with other cells. Thus, detecting and characterizing complexes of cells isolated from tissues and bodily fluids, can provide powerful insights into cell:cell communication events that are missed when studying cells as singlets only.

## Materials and methods

**Key resources table**

| Reagent type (species) or resource | Designation | Source or reference | Identifiers | Additional information |
|---|---|---|---|---|
| Antibody | CCR7-PerCpCy5.5; clone G043H7; mouse monoclonal | Biolegend | Cat# 353220 | 1:50 (4 µl per test) |
| Antibody | CD2-BV421; clone RPA-2.10; mouse monoclonal | Biolegend | Cat# 300229 | 1:66 (3 µl per test) |
| Antibody | CD3-AF700; clone UCHT1; mouse monoclonal | BD pharmigen | Cat# 557943 | 1:66 (3 µl per test) |
| Antibody | CD3-AF488; clone UCHT1; mouse monoclonal | Biolegend | Cat# 300415 | 1:200 (1 µl per test) |
| Antibody | CD4-APCeF780; clone RPA-T4; mouse monoclonal | eBiosciences | Cat# 47-0049-42 | 1:200 (1 µl per test) |
| Antibody | CD5-APCCy7; clone L17F12; mouse monoclonal | Biolegend | Cat# 364009 | 1:66 (3 µl per test) |

*Continued on next page*

*Continued*

| Reagent type (species) or resource | Designation | Source or reference | Identifiers | Additional information |
| --- | --- | --- | --- | --- |
| Antibody | CD7-APC; clone CD7-6B7; mouse monoclonal | Biolegend | Cat# 343107 | 1:66 (3 µl per test) |
| Antibody | CD8a-BV650; clone RPA-T8; mouse monoclonal | Biolegend | Cat# 301042 | 1:200 (1 µl per test) |
| Antibody | CD14-APC; clone 61D3; mouse monoclonal | Tonbo biosciences | Cat# 20–0149 T100 | 1:200 (1 µl per test) |
| Antibody | CD14-AF594; clone HCD14 | Biolegend | Cat# 325630 | 1:200 (1 µl per test) |
| Antibody | CD14-AF647; clone 63D3 | Biolegend | Cat# 367128 | 1:200 (1 µl per test) |
| Antibody | CD14-BV421; clone HCD14 | Biolegend | Cat# 325628 | 1:200 (1 µl per test) |
| Antibody | CD14-PE; clone 61D3; mouse monoclonal | eBioscience | Cat# 12-0149-42 | 1:200 (1 µl per test) |
| Antibody | CD27-BV650; clone O323 | Biolegend | Cat# 302827 | 1:100 (2 µl per test) |
| Antibody | CD33-APC; clone WM53 | Biolegend | Cat# 303407 | 1:200 (1 µl per test) |
| Antibody | CD36-APCCy7; clone 5–271 | Biolegend | Cat# 336213 | 1:100 (2 µl per test) |
| Antibody | CD45-PerCpCy5.5; clone HI30; mouse monoclonal | Tonbo biosciences | Cat# 65–0459 T100 | 1:66 (3 µl per test) |
| Antibody | CD45RA-eF450; clone HI100; mouse monoclonal | eBiosciences | Cat# 48-0458-42 | 1:200 (1 µl per test) |
| Antibody | CD64-AF488; clone 10.1; mouse monoclonal | Biolegend | Cat# 305010 | 1:200 (1 µl per test) |
| Antibody | CD163-PECy7; clone GHI/61; mouse monoclonal | Biolegend | Cat# 333613 | 1:100 (2 µl per test) |
| Antibody | ICAM1(CD54); unconjugated; clone HCD54; mouse monoclonal | Biolegend | Cat# 322704 | 1:40 (5 µl per test) |
| Antibody | LFA1(CD11a); unconjugated; clone TS2/4; mouse monoclonal | Biolegend | Cat# 350602 | 1:40 (5 µl per test) |
| Antibody | LFA1(CD11a/CD18)-AF647; clone m24; mouse monoclonal | Biolegend | Cat# 363412 | 1:40 (5 µl per test) |

## Subjects and samples

Samples from TB uninfected individuals were obtained from the University of California, San Diego Antiviral Research Center clinic (AVRC at UCSD, San Diego) and National Blood Center (NBC), Ministry of Health, Colombo, Sri Lanka, in an anonymous fashion as previously described (*Burel et al., 2017*). Samples from individuals with LTBI were obtained from AVRC at UCSD, San Diego, and the Universidad Peruana Cayetano Heredia (UPCH, Peru). Longitudinal active TB samples were obtained from National Hospital for Respiratory Diseases (NHRD), Welisara, Sri Lanka. Dengue previously infected samples were obtained from healthy adult blood donors from the National Blood Center (NBC), Ministry of Health, Colombo, Sri Lanka, in an anonymous fashion as previously described (*Weiskopf et al., 2013*). Acute dengue fever samples were collected at National Institute of Infectious Diseases, Gothatuwa, Angoda, Sri Lanka and the North Colombo Teaching Hospital, Ragama,

in Colombo, Sri Lanka. Longitudinal Tdap booster vaccination samples and non-vaccinated healthy samples were obtained from healthy adults from San Diego, USA. LTBI status was confirmed in subjects by a positive IFN-γ release assay (IGRA) (QuantiFERON-TB Gold In-Tube, Cellestis or T-SPOT. TB, Oxford Immunotec) and the absence of clinical and radiographic signs of active TB. TB uninfected control subjects were confirmed as IGRA negative. Active Pulmonary TB was defined as those exhibiting symptoms of TB, and are positive by sputum and culture as confirmed by the National Tuberculosis Reference Laboratory (NTRL, Welisara, Sri Lanka). Sputum was further confirmed positive for TB by PCR at Genetech (Sri Lanka). Active TB patients in this study were confirmed negative for HIV, HBV and HCV. Upon enrollment within seven days of starting their anti-TB treatment, active TB patients provided their first blood sample, followed by a second blood sample two months after initial diagnosis. Acute dengue fever and previously infected samples were classified by detection of virus (PCR+) and/or dengue-specific IgM and IgG in the serum. Laboratory parameters such as platelet and leukocyte counts, hematocrit, hemoglobulin, AST, ALT and if applicable an ultrasound examination of the chest and abdomen or an X-ray were used to further diagnose patients with either dengue fever (DF) or dengue hemorrhagic fever (DHF), a more severe form of disease, according to WHO's guidelines. Longitudinal Tdap booster vaccination samples were obtained from individuals vaccinated in childhood, and boosted with the DTP vaccine Tdap (Adacel). Blood samples were collected prior, one day, three days, seven days and fourteen days post boost. Longitudinal healthy samples were obtained from two consecutive bleeds of healthy adults, at seven days apart. For some latent TB and TB negative subjects, leukapheresis was performed instead of a whole blood donation in order to increase the number of PBMC obtained. Samples from Peru were exclusively collected by leukapheresis, whereas 65% of samples from San Diego were collected by leukapheresis (33 out of 51 subjects). No leukapheresis samples were collected in Sri Lanka. We have found no difference in T cell:monocyte complexes frequencies in samples collected by leukapheresis versus whole blood (data not shown), and have thus defined our cohorts based on TB diagnosis status (TB negative, latent TB or active TB), regardless of the blood draw technique. All blood samples were drawn in Lithium or Sodium heparin, except for the analysis of the effect of anti-coagulant (*Figure 2—figure supplement 1B*) where some healthy samples were also collected in EDTA. Clinical sites of Peru and Sri Lanka have been personally trained at La Jolla Institute for Immunology and all three sites follow the same operating procedures and protocols for blood processing. All blood samples were stored at room temperature for up to 12 hr before blood processing with a maximum processing time of three hours. Time of the day for blood draw was aleatory variable ranging from morning to afternoon for each site. For all cohorts, PBMC were obtained by density gradient centrifugation (Ficoll-Hypaque, Amersham Biosciences) according to the manufacturer's instructions. Cells were resuspended to 10 to 50 million cells per mL in FBS (Gemini Bio-Products) containing 10% dimethyl sulfoxide (Sigma) and cryopreserved in liquid nitrogen.

## Magnetic RBC depletion

Magnetic RBC depletion was performed using the EasySep RBC depletion kit (STEMCELL technologies), according to the manufacturer's instructions. Briefly, 500 µl of whole blood was supplemented with 6 mM EDTA (final concentration) and 500 µl PBS + 2% FCS, and transferred into a 5 mL polystyrene round-bottom tube. After adding 25 µl of depletion reagent, the sample was incubated for 5 min on a EasySep magnet, and cell suspension was collected by inverting the magnet in one continuous motion into a new tube. Depletion was repeated once more by adding the same volume of depletion reagent. Cell suspension obtained after the second depletion (depleted of RBC) was directly used for flow cytometry staining.

## Flow cytometry

Surface staining of fresh or frozen PBMC was performed as previously described in *Burel et al. (2017)*. All centrifugations were performed at 600 g for 5 min. For cryopreserved PBMC, cells were quickly thawed by incubating each cryovial at 37°C for 2 min, and cells transferred into 9 ml of cold medium (RPMI 1640 with L-Glutamin and 25 mM Hepes (Omega Scientific), supplemented with 5% human AB serum (GemCell), 1% Penicillin Streptomycin (Gibco) and 1% Glutamax (Gibco)) and 20 U/mL Benzonase Nuclease (Millipore) in a 15 ml conical tube. Cells were centrifuged and resuspended in medium to determine cell concentration and viability using Trypan blue and a hematocytometer.

Cells (1–10 million) were transferred into a 15 ml conical tube, centrifuged, resuspended in 100 µl of PBS containing 10% FBS and incubated for 10 min at 4°C. Cells were then stained with 100 µl of PBS containing fixable viability dye eFluor506 (eBiosciences) and various combinations of the antibodies listed in *Supplementary file 1* for 20 min at room temperature. Each antibody was individually titrated for optimum staining, and dilutions/panels used in the study are available in *Supplementary file 1* . To assess the effect of Fc-receptor blocking on the formation of T cell:monocyte complexes (*Figure 2—figure supplement 1D*), 2 µl of Trustain FcR blocking reagent (BioLegend) was added along with the antibodies. After two washes in staining buffer (PBS containing 0.5% FBS and 2 mM EDTA (pH 8.0), cells were resuspended into 100–500 µl of staining Buffer, transferred into a 5 ml polypropylene FACS tube (BD Biosciences) and stored at 4°C protected from light for up to 4 hr until flow cytometry acquisition. Acquisition was performed on a BD LSR-II cell analyzer (BD Biosciences) or on a BD FACSAria III cell sorter (BD Biosciences). Compensation was realized with single-stained beads (UltraComp eBeads, eBiosciences) in PBS using the same antibody dilution as for the cell staining. Performance of both instruments were checked daily by the flow cytometry core at La Jolla Institute for Immunology with the use of CS and T beads (BD Biosciences), and PMT voltages were manually adjusted for optimum fluorescence detection on each time it was used.

## Imaging flow cytometry

For the visualization of CD3+CD14+ cells, frozen PBMC were thawed and stained with CD3-AF488 and CD14-AF647 (see *Supplementary file 1* for antibody details) as described in the flow cytometry section above. After two washes in PBS, cells were resuspended to $10 \times 10^6$ cells/mL in staining buffer containing 5 µg/mL Hoechst (Invitrogen) and 1 µg/mL 7-AAD (Biolegend) and stored at 4°C protected from light until acquisition. Acquisition was performed with ImageStreamX MkII (Amnis) and INSPIRE software version 200.1.620.0 at 40X magnification and the lowest speed setting. A minimum of 4,000 CD3+CD14+ events in focus were collected. Data analysis was performed using IDEAS version 6.2.183.0.

## Sample preparation for microscopy

For the visualization of LFA1/ICAM1 polarization on T cell:monocyte complexes, frozen PBMC were thawed as described in the flow cytometry section above and resuspended in blocking buffer (2% BSA, 10 mM EGTA, 5 mM EDTA, 0.05% Sodium Azide in 1X PBS) supplemented with 2 µl of Trustain FcR blocking reagent (BioLegend) for 10 min on ice. Antibodies (anti-human CD3-AF488, CD14-BV421, ICAM1-AF568, LFA1-CF633 or LFA1-AF647, see *Supplementary file 1* for antibody details) were added and incubated for 20 min on ice, and then washed twice with staining buffer (PBS containing 0.5% FBS and 2 mM EDTA, pH 8). Cells were fixed with 4% Paraformaldehyde, 0.4% Glutaldehyde, 10 mM EGTA, 5 mM EDTA, 0.05 Sodium Azide, 2% sucrose in PBS for 1 hr on ice, and then washed twice with MACS buffer. Cells were resuspended in 0.5–1 mL of MACS buffer, and kept at 4° C until sorting. Cell sorting was performed on a BD Aria III/Fusion cell sorter (BD Biosciences). CD3+CD14+, CD3+CD14 T cells and CD14 +CD3 monocytes were sorted (see gating strategy *Figure 1—figure supplement 2*) and each separately plated on a well of a µ-Slide 8 Well Glass Bottom chamber (Ibidi) that was freshly coated with poly-L-lysine (0.01%) for 30 min RT before use. For in-house antibody labeling, an Alexa Fluor 568 antibody labeling kit and a Mix-n-Stain CF633 Dye anti-body labeling kit (Sigma) were used according to manufacturer's protocols.

## Microscopy

Airyscan images were taken with a Plan-Apochromat 63x/1.4 Oil DIC M27 objective with a 152 µm sized pinhole with master gain 800 using a Zeiss LSM 880 confocal microscopy equipped with an Airyscan detector (Carl Zeiss). four laser lines at 405, 488, 561, and 633 nm and a filter set for each line were used for taking 20–25 series of z-plane Airyscan confocal images with a step of 0.185 µm or 0.247 µm for each channel. Pixel dwelling time was 2.33 µs and x and y step sizes were 43 nm. 3D-Airyscan processing was performed with the Zen Black 2.3 SP1 program. For some images, Z-plane linear transitional alignment was done by using the Zen Blue 2.5 program. Contrast of images for each fluorophores channel was adjusted based on FMO (Fluorescence minus one) control samples that were prepared and taken on the same day of each experiments. To visualize cell fragments, sorted CD3+CD14mid cells were immobilized using CyGel Sustain (Abcam) according to

manufacturer recommendations. Three-dimensional rendering of cellular fragments (*Figure 2D*) was created in Imaris 9.1 software (Bitplane).

## Disruption of T cell:monocyte complexes

For the RBC lysis condition, immediately after thawing, PBMC were incubated for 10 min at room temperature with 4 mL of 1x RBC lysis reagent (Biolegend) according to the manufacturer's recommendations. After two washes in staining buffer, PBMC were then stained with fixable viability dye, anti-human CD3-AF700 and CD14-PE as described in the flow cytometry section above. For all other conditions, PBMC were first stained and then submitted to one of the following treatments: i) final resuspension in staining buffer at 10 mM EDTA (EDTA 10 mM), ii) vigorous pipetting up and down for 30 s after final resuspension in staining buffer (Pipette Up/Down) or iii) Sonication for 2 min and 30 s at 42 kHz (JSP Ultrasonic Cleaner) after final resuspension in staining buffer (Sonication).

## Bulk memory CD4+ T cell sorting

Frozen PBMC were thawed and stained with fixable viability dye eFluor506 (eBiosciences) and various combinations of the antibodies listed in *Supplementary file 1* as described in the flow cytometry section above. Memory CD4 T cell sorting (see gating strategy *Figure 1—figure supplement 1A*) was performed on a BD Aria III/Fusion cell sorter (BD Biosciences). 100,000 memory CD4+ T cells were sorted into TRIzol LS reagent (Invitrogen) for RNA extraction.

## RNA sequencing and analysis

RNA sequencing and analysis of memory CD4+ T cells from LTBI infected subjects was performed as described in *Picelli et al. (2013)*; *Seumois et al. (2016)* and quantified by qPCR as described previously (*Seumois et al., 2012*). 5 ng of purified total RNA was used for poly(A) mRNA selection, full length reverse-transcription and amplified for 17 cycles, following the smart-seq2 protocol (*Picelli et al., 2013*; *Seumois et al., 2016*). After purification with Ampure XP beads (Ratio 0.8:1, Beckmann Coulter) and quantification (Picogreen assay, Invitrogen), 1 ng of cDNA was used to prepare a Nextera XT sequencing library with the Nextera XT DNA library preparation and index kits (Illumina). Samples were pooled and sequenced using the HiSeq2500 (Illumina) to obtain at least 12 million 50 bp single-end reads per library. The single-end reads that passed Illumina filters were filtered for reads aligning to tRNA, rRNA, and Illumina adapter sequences. The reads were then aligned to UCSC hg19 reference genome using TopHat (v 1.4.1) (*Trapnell et al., 2009*), filtered for low complexity reads, and parsed with SAMtools (*Li et al., 2009*). Read counts to each genomic feature were obtained using HTSeq-count program (v 0.6.0) (*Anders et al., 2015*) using the 'union' option. Raw counts were then imported to R/Bioconductor package DESeq2 (*Love et al., 2014*) to identify differentially expressed genes among samples.

## Data deposition

Sequencing data is accessible online through Gene Expression Omnibus (accession numbers GSE84445 and GSE99373, https://www.ncbi.nlm.nih.gov/geo) and Immport (Study number SDY820, http://www.immport.org). All other data is available in the main text or the supplementary materials.

## Acknowledgements

We thank Dr Chery Kim and all present and past members at the flow cytometry core facility at the La Jolla Institute for Immunology for assistance in cell sorting and technical discussion. We thank Dr Zbigniew Mikulski from the microscopy core at the La Jolla Institute for Immunology for assistance and technical advice on microscopy imaging. We thank Yoav Altman at the Sanford Burnham Prebys flow cytometry core for technical assistance with imaging flow cytometry. We thank Dr Joe Trotter from the R and D Advanced Technology Group at BD Biosciences for useful technical discussion about cytometry instrument fluidic systems. Research reported in this manuscript was supported by the National Institute of Allergy and Infectious Diseases division of the National Institutes of Health under award number U19AI118626, R01AI137681, HHSN272201400045C, P01HL078784, S10OD021831 and S10OD016262. The content is solely the responsibility of the

authors and does not necessarily represent the official views of the National Institutes of Health. Imaging flow cytometry was supported by the James B Pendleton Charitable trust.

## Additional information

### Funding

| Funder | Grant reference number | Author |
|---|---|---|
| National Institute of Allergy and Infectious Diseases | U19AI118626 | Alessandro Sette<br>Bjoern Peters<br>Pandurangan Vijayanand |
| National Institute of Allergy and Infectious Diseases | R01AI137681 | Bjoern Peters |
| National Institutes of Health | HHSN272201400045C | Alessandro Sette |
| National Heart, Lung, and Blood Institute | P01HL078784 | Klaus Ley |

The funders had no role in study design, data collection and interpretation, or the decision to submit the work for publication.

### Author contributions
Julie G Burel, Conceptualization, Investigation, Writing—original draft, Writing—review and editing; Mikhail Pomaznoy, Conceptualization, Investigation, Writing—review and editing; Cecilia S Lindestam Arlehamn, Conceptualization, Resources, Investigation, Writing—review and editing; Daniela Weiskopf, Ricardo da Silva Antunes, Yunmin Jung, Veronique Schulten, Gregory Seumois, Investigation, Writing—review and editing; Mariana Babor, Conceptualization, Writing—review and editing; Jason A Greenbaum, Sunil Premawansa, Gayani Premawansa, Ananda Wijewickrama, Dhammika Vidanagama, Bandu Gunasena, Rashmi Tippalagama, Aruna D deSilva, Robert H Gilman, Mayuko Saito, Randy Taplitz, Klaus Ley, Pandurangan Vijayanand, Resources, Writing—review and editing; Alessandro Sette, Conceptualization, Funding acquisition, Writing—review and editing; Bjoern Peters, Conceptualization, Funding acquisition, Writing—original draft, Writing—review and editing

### Author ORCIDs
Julie G Burel (iD) https://orcid.org/0000-0003-1692-2758
Bjoern Peters (iD) https://orcid.org/0000-0002-8457-6693

### Ethics
Human subjects: Ethical approval to carry out this work is maintained through the La Jolla Institute for Allergy and Immunology Institutional Review Board (protocols VD-143, VD-085, VD-090), the Medical Faculty of the University of Colombo (which served as a National Institutes of Health-approved institutional review board for Genetech, protocols EC-15-094, EC-15-002, EC.15.095), and the Johns Hopkins School of Public Health Institutional Review Board (RHG holds dual appointment at UPCH and JHU; protocol 00003804). All clinical investigations have been conducted according to the principles expressed in the Declaration of Helsinki. All participants, except anonymously recruited blood bank donors in Sri Lanka, provided written informed consent prior to participation in the study.

### Decision letter and Author response
Decision letter https://doi.org/10.7554/eLife.46045.024
Author response https://doi.org/10.7554/eLife.46045.025

## Additional files

**Supplementary files**

• Supplementary file 1. Fluorochrome-conjugated antibodies used in the study.
DOI: https://doi.org/10.7554/eLife.46045.017

• Transparent reporting form
DOI: https://doi.org/10.7554/eLife.46045.018

### Data availability

RNA sequencing data of memory CD4 T cells have been deposited in GEO under accession codes GSE84445 and GSE99373. All data generated or analysed during this study are included in the manuscript and supporting files. An additional source data file has been provided for Figure 1.

The following datasets were generated:

| Author(s) | Year | Dataset title | Dataset URL | Database and Identifier |
|---|---|---|---|---|
| Burel J, Lindenstam C, Seumois G, Fu Z, Greenbaum J, Sette A, Peters B, Vijayanand P | 2016 | Transcriptomic profile of circulating memory T cells can differentiate between latent tuberculosis individuals and healthy controls | https://www.ncbi.nlm.nih.gov/geo/query/acc.cgi?acc=GSE84445 | NCBI Gene Expression Omnibus, GSE84445 |
| Burel J, Lindenstam C, Seumois G, Fu Z, Greenbaum J, Sette A, Peters B, Vijayanand P | 2018 | Transcriptomic profile of circulating memory CD4 T cells can differentiate between latent tuberculosis individuals and healthy controls | https://www.ncbi.nlm.nih.gov/geo/query/acc.cgi?acc=GSE99373 | NCBI Gene Expression Omnibus, GSE99373 |

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
