## [Decision Letter]

Thank you for submitting your article "Circulating T cell-monocyte complexes are markers of immune perturbations" for consideration by *eLife*. Your article has been reviewed by three peer reviewers, including Jos W van der Meer as the Reviewing Editor and Reviewer #1, and the evaluation has been overseen by Satyajit Rath as the Senior Editor. The following individual involved in review of your submission has agreed to reveal their identity: Claude Lambert (Reviewer #2).

The reviewers have discussed the reviews with one another and the Reviewing Editor has drafted this decision to help you prepare a revised submission.

Summary:

The authors report on the existence of monocytes and T-lymphocytes that are found complexed using the cell sorter. The interaction does not seem to occur as a classical immunological synapse, but rather as LFA1 ICAM1 interaction.

They provide evidence that this feature varies under different clinical conditions and that the most prevalent T-cell subset in the complex also varies. They also manage to define a new, convenient parameter based on biochemistry Ka calculation (although this parameter may not be easy to use in routine diagnosis). All together, the reviewers agree that this study is of high interest.

The major question however is whether there is convincing proof in the paper that this is indeed occurring in vivo (and thus in the blood stream) and it is not an ex-vivo artefact.

Essential revisions:

1) Given the critical issue whether this is an in-vivo or an ex-vivo phenomenon, it is amazing that the pre-analytical procedures are so poorly (i.e., not at all) described. What time of the day was the bleeding performed? How many were leukapheresis samples? Why leukapheresis? Which anticoagulants were used? What tubes were used? How long were the tubes at room temperature waiting to be processed? How long was the total procedure?

Probably the conditions were different in San Diego, Peru and Sri Lanka. These should also be reported.

2) In line with the criticism under point 1, data on the intra-individual variation and the effect of cryopreservation are lacking. It would be good to do a carefully controlled experiment in perhaps 3 volunteers, in which these individuals are bled on two separate days. Cells are processed immediately for FACS analysis and are cryopreserved for later testing. On the second occasion the cryopreserved cells are used as well as freshly processed cells from the second bleed. Ideally, the role of different anticoagulants (EDTA, heparin, citrate) could be assessed in the same setting.

3) The exact experimental conditions of the magnetic depletion (subsection “T cell:monocyte complexes are not the result of cryopreservation or PBMC isolation”) are also lacking in the Materials and methods section.

4) In the Materials and methods section a very poor description of the FCM procedures is given. No data about panel compositions, PMT setting, compensation, titrations etc. as well as composition of buffers are described. This is essential to check the value of data.

5) The stability of the complexed cells is not mentioned. Can the complexes be disrupted by pipetting or by (mild) sonification?

6) Conversely, do more complexes occur when a small dose of LPS or a bioactive cytokine is added in vivo for say 2-3 hours?

7) Although authors give arguments (also in Discussion) to underline their hypothesis of T cell-monocyte complexes, it cannot be excludes that the binding of the conjugates for immunophenotyping (the MoAb-fluorochrome) can be the result of the complex formation (as is known in studies of monocytes),. Fc-receptor binding may occur and result in capping of the adhesion molecules/receptors or non-specific binding as such. This should be ruled out and clarified.

8) The data show that cell-cell interaction can end in exchange of cellular material (in this case, CD14+ monocyte membrane fragments on T cells) This could certainly have something to do with "trogocytosis", previously described by Hudrisier and co-workers (Daubeuf et al., 2010) This should be discussed.

9) Supplementary file 1 lists a number of MoAbs used in this study, but the gating when using these antibodies is not adequately shown. Furthermore, the reviewers would like to have additional information on antibodies, such as CD45 (for gating); CD36, CD64, CD33, CD163, CD68 (monocyte identification, CD14 is also expressed dimly on neutrophils); CD2, CD5, CD7, CD27 (T cells) etc.

10) The definition of memory T cells is not correct and the need in this study is not clearly documented.

11) Gating procedures do not seem to be correctly performed and will cause contamination of gates by non-target cells. No use of CD45/SS (eliminate debris, RBCs and platelets) and sequential back gating (to check the correct population of interest) is applied in the various figures. By example CD4 is dimly expressed on monocytes and should be excluded.

12) Ka is used to correct for variation in T cells and monocytes during immune processes, however the absolute amount of T-Mono complexes should be given and not only the frequencies. This may also explain why CD4+CD8+ and CD4-CD8, which represent relatively small populations, have high frequencies.

13) As such the higher amount of Ka frequencies in CD4+CD8+ (double positive) and CD4-CD8- (double negative) T cells (Figure 3—figure supplement 3 4) are not discussed and explained.

---

## [Author Response]

Essential revisions:1) Given the critical issue whether this is an in-vivo or an ex-vivo phenomenon, it is amazing that the pre-analytical procedures are so poorly (i.e., not at all) described. What time of the day was the bleeding performed? How many were leukapheresis samples? Why leukapheresis? Which anticoagulants were used? What tubes were used? How long were the tubes at room temperature waiting to be processed? How long was the total procedure? Probably the conditions were different in San Diego, Peru and Sri Lanka. These should also be reported.

We have now expanded on the conditions for blood drawing and processing in the Materials and methods of the revised manuscript (Subjects and Samples section). Some individuals from San Diego and all from Peru donated by leukapheresis since we needed a large number of cells for multiple studies under the same scope of understanding TB immunity. Leukapheresis typically results in 10x more PBMC as compared to a full unit blood draw (450ml=450 million PBMC, whereas leukapheresis typically yield ~5 billion PBMC).

2) In line with the criticism under point 1, data on the intra-individual variation and the effect of cryopreservation are lacking. It would be good to do a carefully controlled experiment in perhaps 3 volunteers, in which these individuals are bled on two separate days. Cells are processed immediately for FACS analysis and are cryopreserved for later testing. On the second occasion the cryopreserved cells are used as well as freshly processed cells from the second bleed. Ideally, the role of different anticoagulants (EDTA, heparin, citrate) could be assessed in the same setting.

As requested, we have now incorporated the results from a longitudinal study (two blood draws, one week apart) of a cohort of 5 healthy donors, where we measured T cell:monocyte complexes in both fresh and frozen PBMC obtained from each blood draw. Strikingly, whereas the frequency of T cell:monocyte complexes was variable between individuals, it was highly stable over time within each individual, and this was true for both fresh and frozen PBMC (non-parametric spearman correlation r=1 and r=0.9 for fresh and frozen PBMC, respectively). In parallel, we have also compared the effect of the anti-coagulant used for blood collection, and we have found no differences in the frequency of T cell:monocyte complexes between blood samples drawn with either EDTA or Heparin, in both fresh and frozen PBMC. Results have been reported in Figure 2—figure supplement 1B for the anti-coagulant comparison, and Figure 2—figure supplement 1C for the longitudinal bleeds.

3) The exact experimental conditions of the magnetic depletion (subsection “T cell:monocyte complexes are not the result of cryopreservation or PBMC isolation”) are also lacking in the Materials and methods section.

We apologize for this omission. Full details on the experimental conditions for the red blood cells magnetic depletion have been included in the revised manuscript (Materials and methods section “Magnetic RBC depletion”).

4) In the Materials and methods section a very poor description of the FCM procedures is given. No data about panel compositions, PMT setting, compensation, titrations etc. as well as composition of buffers are described. This is essential to check the value of data.

We apologize for the light description of our procedures despite being of clear importance for other scientists to reproduce our results. Accordingly, we have now included a full detailed description of the flow cytometry procedures in the Materials and methods section of the revised manuscript (”Flow cytometry” section, and Supplementary file 1) including Ab panels/dilutions, centrifugation parameters, composition of staining buffers and acquisition parameters.

5) The stability of the complexed cells is not mentioned. Can the complexes be disrupted by pipetting or by (mild) sonification?

As requested, we investigated various in vitro conditions in which we attempted to break down the cell-cell interaction in T cell:monocyte complexes. We assessed the effect of 4 ‘disruptive’ conditions on T cell:monocyte complexes frequency in frozen PBMC samples from 3 independent healthy individuals. In all 3 individuals tested, the frequency of T cell:monocyte complexes was reduced after vigorous pipetting up/down or mild sonication, the strongest ‘destructive’ effect observed with the later condition. Incubation with RBC lysis buffer disrupted the T cell:monocyte complexes in 1 out of 3 individuals, while addition of high concentration of anti-chelating agent EDTA had no effect on their frequency. These results are reported in Figure 2—figure supplement 3 of the revised manuscript.

*6) Conversely, do more complexes occur when a small dose of LPS or a bioactive cytokine is added* in vivo *for say 2-3 hours?*

During the course of this project, we have on several occasions attempted to promote the formation of T cell:monocyte complexes in vitro, by incubating PBMC with various concentrations of highly antigenic stimuli including LPS, SEB or PHA, as well as live pathogens such as live BCG. Our length of stimulation was variable from as little as 2h, to as long as 5 days. In none of these attempts did the frequency of T cell:monocyte complexes increase after stimulation. If anything, we saw a trend for a lower frequency of T cell:monocyte complexes over time, in particular in the unstimulated samples, suggesting the process of cell culture in itself might impact their frequency. Since monocytes (and thus likely T cell:monocyte complexes) are known to stick to plastic from plates used for cell culture, these experiments are not conclusive. While we agree that it would be of interest to set up an in vitro model that promote the formation of T cell:monocyte complexes and allows to reliably quantifying them, potentially through the use of non-sticky cell culture consumables or post culture detaching protocols specific to monocytes, this will require more carefully designed experiments, which goes beyond the scope of this current manuscript revision.. Nevertheless, we have included the following sentence in the Results section in the revised manuscript: “Conversely, we tried to promote their in vitro formation by stimulating PBMC for two hours to several days, with various concentrations of highly antigenic stimuli (such as LPS, PHA, SEB or live BCG) without success, which might be due to issues with monocytes attaching to the culture plates (data not shown).”

7) Although authors give arguments (also in Discussion) to underline their hypothesis of T cell-monocyte complexes, it cannot be excludes that the binding of the conjugates for immunophenotyping (the MoAb-fluorochrome) can be the result of the complex formation (as is known in studies of monocytes),. Fc-receptor binding may occur and result in capping of the adhesion molecules/receptors or non-specific binding as such. This should be ruled out and clarified.

To rule out that non-specific binding of antibody conjugates used for flow cytometry staining could be the result of complex formation, we compared the frequency of T cell:monocyte complexes obtained from frozen PBMC in the presence or absence of a Fc-receptor binding reagent. No difference in T cell:monocyte complexes frequencies was observed when blocking Fc-receptor binding of conjugated antibodies. The results are presented in Figure 2—figure supplement 1D in the revised manuscript.

8) The data show that cell-cell interaction can end in exchange of cellular material (in this case, CD14+ monocyte membrane fragments on T cells) This could certainly have something to do with "trogocytosis", previously described by Hudrisier and co-workers (Daubeuf et al., 2010) This should be discussed.

We agree with the reviewers that some CD3+CD14mid cells might well be the result of T cells that have acquired some monocyte membrane fragments after interaction. We have thus incorporated the concept of trogocytosis in the third paragraph of the Discussion.

9) Supplementary file 1 lists a number of MoAbs used in this study, but the gating when using these antibodies is not adequately shown. Furthermore, the reviewers would like to have additional information on antibodies, such as CD45 (for gating); CD36, CD64, CD33, CD163, CD68 (monocyte identification, CD14 is also expressed dimly on neutrophils); CD2, CD5, CD7, CD27 (T cells) etc.

An additional column has now been incorporated in Supplementary file 1, specifying for each MoAb, the figure in which it was used to generate the data. As for the gating, the gating strategy to identify T cell:monocyte complexes was consistent throughout the manuscript, as represented in Figure 1—figure supplement 2. Whenever another gating strategy was used (for instance to detail CD3+CD14hi and CD3+CD14mid, or when CD45-SSC gating was used), this is clearly mentioned in both text and figure legends, and the alternative gating strategy is available in supplemental figures. To improve clarity, we have now included the following sentence in the Results section of the revised manuscript: “For the remaining of the manuscript, we refer to T cell:monocyte complexes as the CD3+CD14+ population gated from live singlets cells, as represented in Figure 1—figure supplement 2.” This was also specified in each figure’s legend.

The use of CD45 staining for better identification of T cell:monocytes complexes has been addressed and discussed later in essential revision point #11. For better identification of the nature of the T cell and monocyte present in the complexes, we analyzed the expression of various canonical markers for monocytes and T cells within T cell:monocyte complexes from healthy individuals. Monocytes in a complex (CD14+CD3+) vs. not in a complex (CD14+CD3-) had similar expression for monocyte canonical markers: CD33, CD36, CD64 and CD163. Similarly, no difference in the levels of expression of T cell canonical markers: CD2, CD5, CD7 and CD27, was observed between T cells in a complex (CD3+CD14+) vs. not in a complex (CD3+CD14-). Additionally, both CD4 and CD8 T cell subsets could be found in association with a monocyte, as well as naïve and memory phenotypes. Thus, there was no obvious T cell or monocyte marker that could differentiate between monocytes or T cells present in a complex vs. not in a complex, suggesting the ability to form complexes is a general property of all CD14+ monocytes and T cell subsets. Accordingly, these results have now been incorporated in Figures 3C-F of the revised manuscript.

10) The definition of memory T cells is not correct and the need in this study is not clearly documented.

The gating definition of memory CD4 T cells is following the standard cell ontology (including CD45RA-CCR7+ central memory T cells, CD45RA-CCR7- effector memory T cells and CD45RA+CCR7- effector memory T cells re-expressing CD45RA) and has been used by us and several others groups (Campion et al. PMID: 24958850, Becattini et al. PMID: 25477212, Tian et al. PMID: 29133794, Burel et al. PMID: 29602771). We acknowledge that recent findings suggest that this definition will exclude stem-cell like memory T cells, which present a ‘naïve’ CD45RA+CCR7+ phenotype. Considering that they represent a very small fraction of total CD4 T cells in the blood, we think it is unlikely that their inclusion would have changed the results obtained here. Additionally, we are reporting on our previous study of memory CD4 T cells in the context of latent TB infection as our first observation of the existence of CD3+CD14+ cells. This observation arose from the analysis of their transcriptomic profile, which contained some monocyte-associated genes consistently across all subjects analyzed, regardless of their TB infection status. The focus on this primary study was to define novel gene signatures that can distinguish between latent TB and TB negative subjects, and gain insights into the phenotype of Mtb-specific CD4 T cells. Because all Mtb-reactive CD4 T cells were expected to fall into the memory CD4 T cell compartment, this particular cell population was sorted to analyze their transcriptomic profile. In this new manuscript, we are solely referring to this dataset (and thus to memory CD4 T cells) as the starting point of our study, to explain our scientific rationale behind looking further into CD3+CD14+ cells. However, we think similar results would have been obtained by looking into other T cell subsets such as naïve CD4 T cells, or CD8 T cells. Indeed, we have found that both CD4 and CD8 T cells are able to form complexes with monocytes, and present both naïve and memory phenotypes (see essential revision point #9, Figure 3E-F). Accordingly, we have clarified the rationale behind looking at memory CD4 T cells in the Discussion (second paragraph) of the revised manuscript.

11) Gating procedures do not seem to be correctly performed and will cause contamination of gates by non-target cells. No use of CD45/SS (eliminate debris, RBCs and platelets) and sequential back gating (to check the correct population of interest) is applied in the various figures. By example CD4 is dimly expressed on monocytes and should be excluded.

We found that the frequency of T cell:monocyte complexes was not affected by applying an initial CD45-SSC filtering to our gating strategy. Similarly, there was no difference between the density plots of CD3+ cells, CD14+ cells and CD3+CD14+ cells for any combination of A, H and W for both FSC and SSC parameters. Thus, our conclusion is that we could not find any parameter from conventional flow cytometry that could differentiate between singlets and T cell:monocyte complexes. These results have been incorporated in Figure 3A (for FSC/SSC parameters) and Figure 3B (for CD45-SSC gating) in the revised manuscript.

As for the expression of CD4 in monocytes, we indeed observe a higher MFI for CD4 in monocytes (and in T cell:monocyte complexes) than for CD8+ T cells, but this does not preclude the breakdown into CD4/CD8 subsets, as seen in Figure 3E. Furthermore, CD4 is not being used in the upstream gating strategy to identify T cell:monocyte complexes, which is solely based on the expression of CD3 and CD14 (Figure 1—figure supplement 2). Thus, CD4 expression cannot impact T cell:monocyte frequencies.

12) Ka is used to correct for variation in T cells and monocytes during immune processes, however the absolute amount of T-Mono complexes should be given and not only the frequencies. This may also explain why CD4+CD8+ and CD4-CD8-, which represent relatively small populations, have high frequencies.

We agree it would be of great interest to compare the absolute counts in complexes between our different disease cohorts. However, since no full blood counts were performed on the blood draws from any of these donors, it is unfortunately not possible to get absolute counts for any cell population, including T cell:monocyte complexes. Frequencies of T cell:monocyte complexes as percent of live cells are already included in Figure 4—figure supplement 4. Indeed, as predicted by the reviewers, we see a lower frequency of DPOS and DNEG T cell:monocyte complexes compared to CD4+ or CD8+ T cell:monocyte complexes, likely due to their low frequency within T cells (Figure 4—figure supplement 4D). However, interestingly their Ka is much higher than with single positive CD4 and CD8 T cells, especially for DNEG T cells (Figure 4F). This suggests that, despite of being in low abundance, DPOS and DNEG have a higher likelihood to form a complex with a monocyte. This information would have been missed by looking at frequencies only. This is the reason why we think the Ka is a more accurate parameter to measure the likelihood of in vivo formation of T cell:monocyte complexes (and correct for aleatory/random association). We acknowledge that this point might not have come across clearly in our first manuscript version and have thus emphasized it in the Results section “The frequency of T cell:monocyte complexes varies in the context of diverse immune perturbations” and in the fifth paragraph of the Discussion section.

13) As such the higher amount of Ka frequencies in CD4+CD8+ (double positive) and CD4-CD8- (double negative) T cells (Figure 3—figure supplement 3 and 4) are not discussed and explained.

A discussion on the enrichment for DNEG cells in T cell:monocyte complexes was included in the original manuscript submission. The enrichment for DPOS T cell:monocytes complexes was not mentioned in our original submission and is now discussed in the sixth paragraph of the Discussion.